# Gasdermin-D-dependent IL-1α release from microglia promotes protective immunity during chronic *Toxoplasma gondii* infection

Samantha J. Batista[1], Katherine M. Still[1], David Johanson[1], Jeremy A. Thompson[1], Carleigh A. O'Brien[1], John R. Lukens [1] & Tajie H. Harris [1✉]

Microglia, resident immune cells of the CNS, are thought to defend against infections. *Toxoplasma gondii* is an opportunistic infection that can cause severe neurological disease. Here we report that during *T. gondii* infection a strong NF-κB and inflammatory cytokine transcriptional signature is overrepresented in blood-derived macrophages versus microglia. Interestingly, IL-1α is enriched in microglia and IL-1β in macrophages. We find that mice lacking IL-1R1 or IL-1α, but not IL-1β, have impaired parasite control and immune cell infiltration within the brain. Further, we show that microglia, not peripheral myeloid cells, release IL-1α ex vivo. Finally, we show that ex vivo IL-1α release is gasdermin-D dependent, and that gasdermin-D and caspase-1/11 deficient mice show deficits in brain inflammation and parasite control. These results demonstrate that microglia and macrophages are differently equipped to propagate inflammation, and that in chronic *T. gondii* infection, microglia can release the alarmin IL-1α, promoting neuroinflammation and parasite control.

[1] Center for Brain Immunology and Glia, Department of Neuroscience, University of Virginia, Charlottesville, VA 22908, USA. ✉email: tajieharris@virginia.edu

Brain infections cause significant morbidity and mortality worldwide. Many of these pathogens persist in a chronic latent form in the brain and require constant immune pressure to prevent symptomatic disease. As the resident macrophage, microglia are widely assumed to control CNS infections, but in many contexts their specific role remains poorly understood. One CNS-tropic pathogen is *Toxoplasma gondii*, a eukaryotic parasite with a broad host range that infects a large portion of the human population[1–5]. *T. gondii* establishes chronic infections by encysting in immune privileged organs, including the brain[6,7]. Without sufficient immune pressure, a fatal neurological manifestation of this disease toxoplasmic encephalitis can occur[2,4,5].

Studies done in mice, a natural host of this parasite, have elucidated many aspects of the immune response that are essential for maintaining control of the parasite during chronic stages of infection. Whole-brain RNA-sequencing has shown that infected mice differ in gene expression from uninfected mice, and exhibit enrichment for immune and inflammation-related pathways[8]. Brain resident cells such as astrocytes have been demonstrated to produce chemokines and cytokines in the brain that promote inflammation and parasite control, as well as to be able to directly kill parasites in vitro[9–16]. T cells infiltrate the brain during infection, and T cell-derived IFN-γ is essential to the control of chronic infection[17–19]. IFN-γ acts on target cells to induce an anti-parasitic state, allowing for the destruction of the parasite through a number of mechanisms including the recruitment of immunity-related GTPases (IRGs) and guanylate binding proteins (GBPs) to the parasitophorous vacuole, as well as the production of nitric oxide (NO)[20–25]. Large numbers of monocytes and monocyte-derived macrophages, a target population for IFN-γ signaling[21], are recruited into the brain parenchyma, and these cells are also necessary for maintaining control of the parasite and host survival[26]. Though microglia occupy the same environment as these cells in the infected brain, and have an activated morphology, their role in chronic *T. gondii* infection has not been fully elucidated. In culture systems, primed microglia have been shown to limit parasite replication[27–29]. In vitro and in vivo studies have also shown that microglia and infiltrating macrophages can produce chemokines and cytokines in the brain during infection and can also display migratory behavior[9,14,30–34]. Whether microglia and recruited macrophages respond in similar ways to brain infection is still being explored, and the new tools we use allow us to investigate non-overlapping functions specific to microglia or macrophages in the brain during chronic infection.

IL-1 molecules include two main cytokines: IL-1α and IL-1β. IL-1α can function as a canonical alarmin, which is a pre-stored molecule that does not require processing and can be released upon cell death or damage, making it an ideal candidate for an early initiator of inflammation[35]. In contrast, IL-1β is produced first as a pro-form that requires cleavage by caspase-1 to be biologically active, rendering IL-1β dependent on the inflammasome as a platform for caspase-1 activation[36,37]. Both of these cytokines signal through the same receptor (IL-1R), a heterodimer of IL-1R1 and IL-1RAcP, with similar affinity[38]. They also require a loss of membrane integrity to be released. Caspase-mediated cleavage of gasdermin molecules has been identified as a major pathway leading to pore formation and IL-1 release.

The role of IL-1β and inflammasome pathways in *T. gondii* infection has been studied in vitro as well as in rodent models of acute infection. In sum, these studies suggest roles for IL-1β, IL-18, IL-1R, NLRP1 and/or NLPR3 inflammasome sensors, the inflammasome adaptor protein ASC, and inflammatory caspases-1 and -11[39–42]. However, the role of IL-1 signaling in the brain during chronic infection has not been addressed.

Here, we show that though they are present in the same tissue microenvironment in the brain during *T. gondii* infection, monocyte-derived macrophages have a stronger NF-κB signature than brain-resident microglia. Interestingly, we also find that these two cell types display differences in the expression of IL-1 molecules. IL-1 signaling contributes to parasite control and the recruitment of immune cells to the brain. We find IL-1R1 expression predominantly on blood vasculature in the brain, and observe IL-1-dependent activation of the vasculature during infection. The pro-inflammatory effect of IL-1 signaling is mediated via the alarmin IL-1α, not IL-1β. We show that microglia, not infiltrating macrophages, release IL-1α ex vivo in an infection-dependent and gasdermin-D-dependent manner. We propose that one specific function of microglia during *T. gondii* infection is to release the alarmin IL-1α to promote neuroinflammation and parasite control.

## Results

**Microglia lack an NF-κB signature in the infected brain.** *T. gondii* infection results in robust, sustained brain inflammation that is necessary for parasite control. Blood-derived monocytes have been demonstrated to be important for host survival during infection[26]. Microglia have also been demonstrated to possess some potentially anti-parasitic functions[27–31], but we are interested in investigating whether microglia perform specific, non-overlapping functions distinct from infiltrating macrophages. Previous work from our lab has observed that while blood-derived monocytes and macrophages express high levels of the nitric oxide-generating enzyme iNOS in the brain during *T. gondii* infection, microglia lack this anti-parasitic molecule[43]. We hypothesize that even though they are in the same tissue microenvironment, microglia are unable to respond to the infection in the same way as infiltrating macrophages. We use a CX3CR1$^{\text{Cre-ERT2}}$ × ZsGreen$^{\text{fl/stop/fl}}$ mouse line that has been previously described as a microglia reporter line[44]. Reporter mice were treated with tamoxifen to induce ZsGreen expression and rested for 4 weeks after tamoxifen injection to ensure the turnover of peripheral CX3CR1-expressing cells. We have consistently used this mouse line in our lab to label over 98% of microglia in the brain. Perivascular macrophages are also labeled by this method, but are not purified by our isolation protocol as evidenced by a lack of CD206$^+$ cells. Following infection, FACS was used to sort out CD45$^+$CD11b$^+$ ZsGreen$^+$ microglia and ZsGreen$^-$ blood-derived myeloid cells from brains of infected mice for RNA sequencing analysis (Fig. 1a).

Analysis of differentially expressed genes shows that these two cell populations segregate clearly from each other, confirming that they are fundamentally different cell types (Fig. 1b). Analysis of pathway enrichment displayed a striking lack of an inflammatory signature in microglia compared to macrophages (Fig. 1c), and we further show a selection of genes that were differentially expressed, showing a clear enrichment for inflammation-associated genes in the macrophage population (Fig. 1d). Interestingly, an NF-κB signature seemed to be one factor differentiating the macrophages from the microglia (Fig. 1c, d). A difference in expression of NF-κB genes could provide the basis for functional differences between microglia and macrophages and their ability to respond to the infection. We validated this at the protein level, showing that in brain sections from infected microglia reporter mice, both RelA and Rel were distinctly absent from ZsGreen$^+$ microglia (Fig. 1e, f) but these molecules were present in ZsGreen$^-$Iba1$^+$ macrophages (Fig. 1g, h). This suggests that some aspects of microglia identity may inhibit the upregulation of a certain inflammatory signature during infection, including a strong NF-κB response.

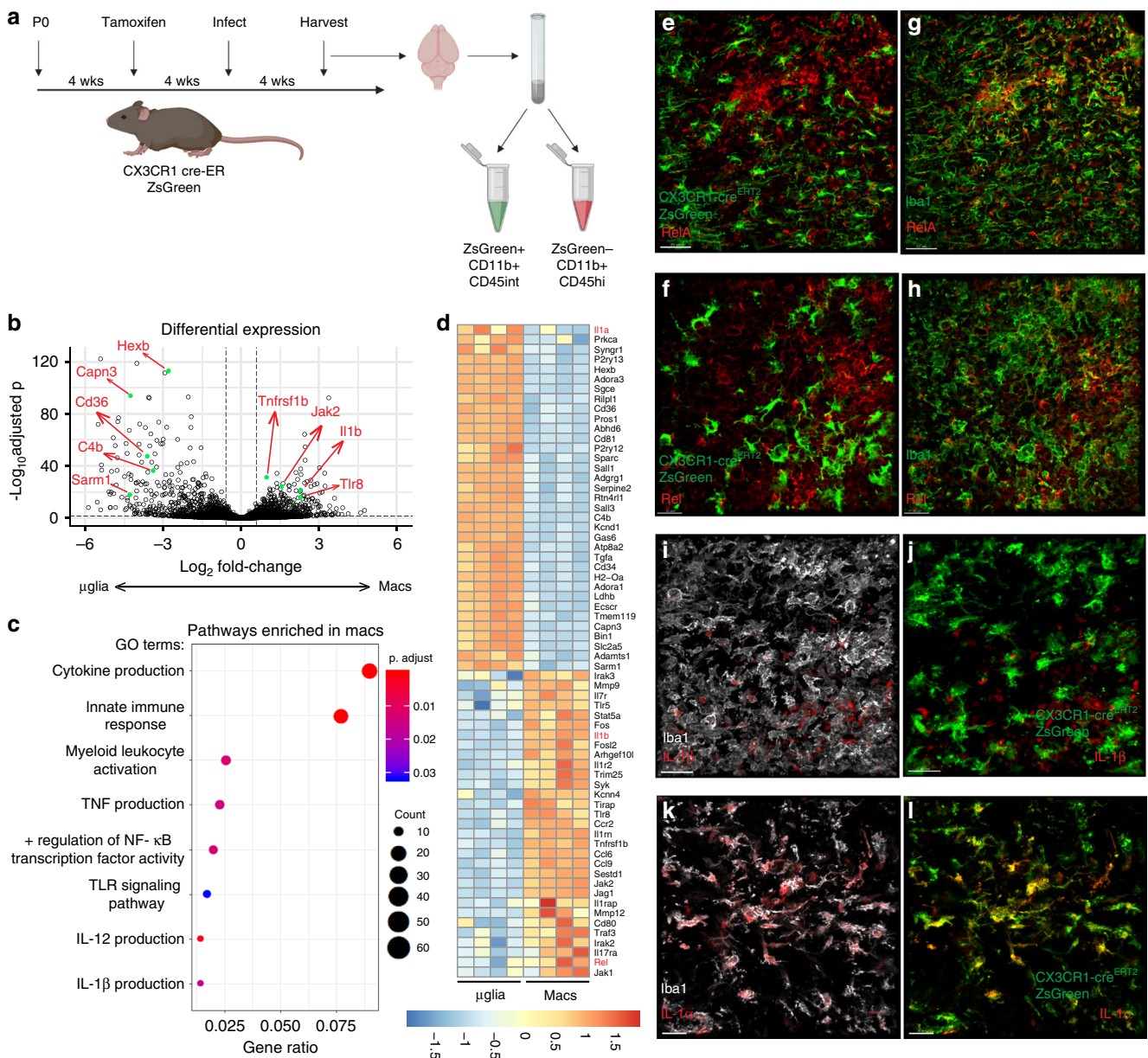

**Fig. 1 Microglia lack an NF-κB signature in the infected brain. a–d** Chronically infected CX₃R₁^Cre-ERT2 × ZsGreen^fl/stop/fl mice were sacrificed and brains were harvested and processed for flow cytometry (*n* = 4 mice). Samples were run on a BD Aria, gated on live/singlets/CD45⁺/CD11b⁺ from which ZsGreen⁺ and ZsGreen⁻ populations were gated and sorted. Sorted cell populations were subjected to RNA sequencing. **a** Experimental setup. **b** Differential abundance testing was performed and results were plotted in R to produce a volcano plot showing differentially expressed genes between microglia and macrophage populations. Example genes are labeled in red corresponding to green dots. **c** GO terms statistically over-represented in macrophages compared to microglia were generated and a selection of significantly enriched pathways of interest were plotted using R. **d** Significantly differentially expressed genes between the two cell populations were selected based on interest and plotted in a heatmap using complete-linkage clustering of a euclidean distance matrix of all samples. **e–l** Representative images of brain sections from chronically infected CX₃R₁^Cre-ERT2 × ZsGreen^fl/stop/fl mice. Experiment was performed twice. ZsGreen is shown in green (**e**, **f**, **j**, **l**) and sections were stained for Iba1 (green, **g**, **h**; gray, **i**, **k**), RelA (**e**, **g** red), Rel (**f**, **h** red), IL-1β (**i**, **j** red), and IL-1α (**k**, **l** red). Scale bars indicate 30 μm.

## IL-1 genes are differentially expressed by microglia.

It was observed that the IL-1 cytokines segregated differently between these populations. IL-1α was enriched in the microglia, while IL-1β was enriched in the macrophages (Fig. 1d). This suggests that these two cell types may be differently equipped to propagate innate inflammatory signals. The lack of microglia expression of pro-IL-1β was validated at the protein level in sections from infected microglia reporter mice, where it is localized in ZsGreen⁻Iba1⁺ cells (Fig. 1i, j). IL-1α is localized generally in Iba1⁺ cells (Fig. 1k), including within ZsGreen⁺ microglia

(Fig. 1l). These results were further confirmed using flow cytometry analysis in both WT and microglia reporter mice. IL-1α protein is present in the brain prior to infection where it is found in microglia as defined by ZsGreen⁺ or CD11b⁺CD45^int (Supplementary Fig. 1a, b, d). During chronic infection, it is expressed by both ZsGreen⁺ microglia and ZsGreen⁻ myeloid cells (Supplementary Fig. 1b) also defined by CD11b⁺CD45^int and CD45^hi (Supplementary Fig. 1f). IL-1β was not detected in uninfected brains, but was present in the brain during chronic *T. gondii* infection (Supplementary Fig. 1bf). During chronic infection,

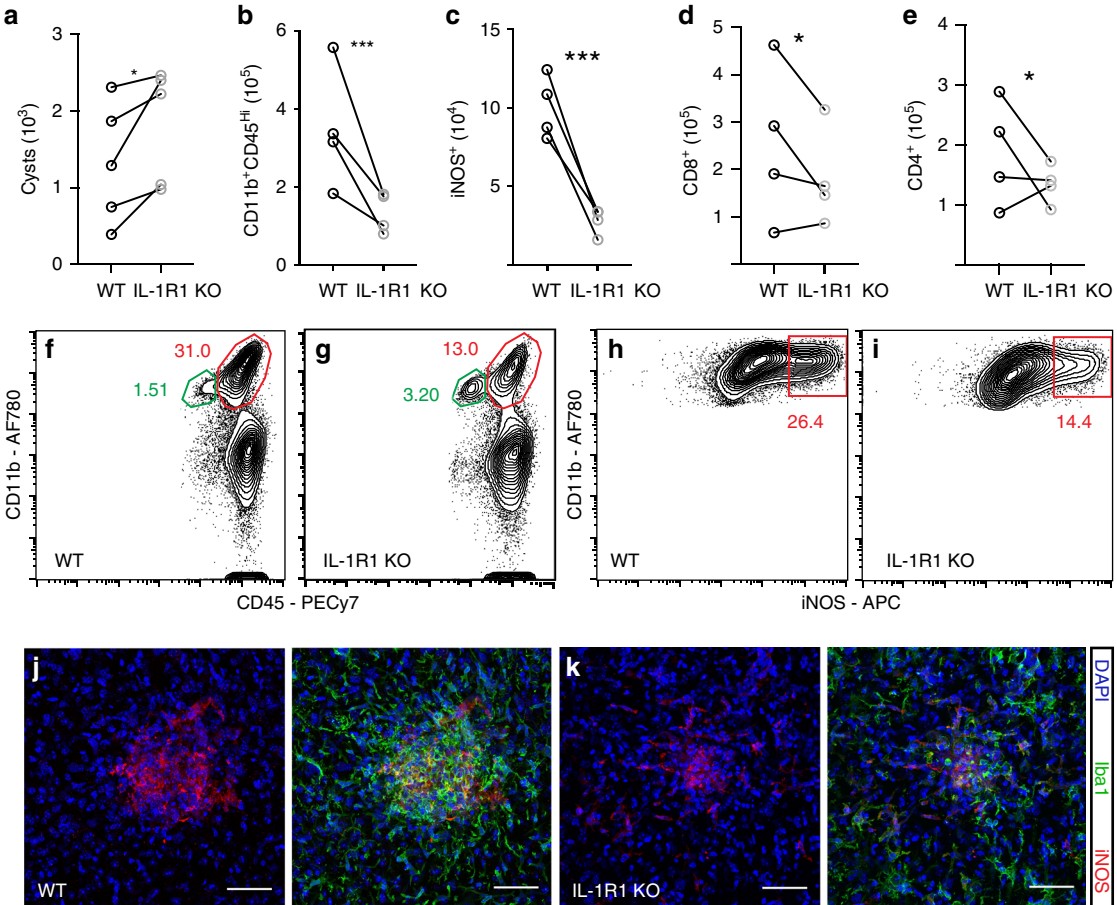

**Fig. 2 IL-1R1 KO mice have an impaired immune response to infection.** WT C57B6/J or IL-1R1 KO mice were infected i.p. with 10 cysts of the Me49 strain of *T. gondii*. 6 weeks p.i. brains were harvested and homogenized. **a** Cyst burden per brain was determined by counting cysts in brain homogenate on a light microscope. Paired averages from five experiments are shown, and statistics were performed using a randomized block ANOVA (two way). $p = 0.048$ ($n = 35$ mice) **b-i** Brains from the same mice were processed to achieve a single cell suspension and analyzed by flow cytometry. Data compiled from four experiments; statistics were performed using a randomized block ANOVA (two way). ($n = 33$ mice). **b** Blood-derived myeloid cells were defined as CD11b$^+$CD45$^{hi}$, cells were pre-gated on singlets/live/CD45$^+$/CD11c$^-$, $p = 2.07 \times 10^{-7}$, representative flow plots are shown in (**f**, **g**). **c** The number of iNOS$^+$ cells per brain were calculated, pre-gated on singlets/live/CD45$^+$/CD11c$^-$/CD11b$^+$CD45$^{hi}$, $p = 3.99 \times 10^{-7}$, representative flow plots are shown in (**h**, **i**). **d**, **e** CD8$^+$ ($p = 0.015$) and CD4$^+$ ($p = 0.035$) T cell numbers were calculated, pre-gated on singlets/live/CD3$^+$. **j**, **k** Representative confocal images of focal areas of inflammation in chronically infected brains of WT (**j**) and IL-1R1 KO (**k**) mice. Scale bars indicate 50 μm. Source data (**a-e**) are provided as a Source data file.

pro-IL-1β and was not significantly expressed by ZsGreen$^+$ cells, but was rather seen in ZsGreen$^-$ myeloid cells (Supplementary Fig. 1c) also defined as CD11b$^+$CD45$^{hi}$ cells (Supplementary Fig. 1g). It was also apparent that while ZsGreen$^-$ blood-derived myeloid cells can express both IL-1α and pro-IL-1β, very few ZsGreen$^+$ microglia were double positive (Supplementary Fig. 1d). These data suggest that microglia and macrophages may play different roles in an IL-1 response. Thus, we aimed to investigate the potential importance of an IL-1 response in *T. gondii* infection.

**IL-1R1 KO mice have an impaired immune response to infection**. To determine if IL-1 signaling plays a role in chronic *T. gondii* infection, we infected mice lacking the IL-1 receptor (IL-1R1), which is bound by both IL-1α and IL-1β. Six weeks post-infection (p.i.) IL-1R1 KO mice displayed an increase in parasite cyst burden in the brain (Fig. 2a). An increase in parasite burden is often due to impaired immune responses. Indeed, IL-1R1 KO mice also have a decrease in the number of CD11b$^+$CD45$^{hi}$ cells of the monocyte/macrophage lineage in the brain during chronic

infection (Fig. 2b, f, g). Microglia typically express intermediate levels of CD45 compared to the high levels expressed by blood-derived myeloid cells, thus we use this marker as a proxy to define these populations by flow cytometry[45]. The cells we defined as infiltrating monocyte/macrophages are also Ly6G$^-$, CD11c$^-$, and Ly6C$^+$. Infiltrating myeloid cells are important producers of nitric oxide, a key anti-parasitic molecule, and thus we assessed their expression of inducible nitric oxide synthase (iNOS). IL-1R1 KO mice had significantly decreased expression of iNOS in the brain compared to WT mice (Fig. 2c, h, i), which was observed specifically in focal areas of inflammation (Fig. 2j, k). Of note, though there were decreases in CD4$^+$ and CD8$^+$ T cells (Fig. 2d, e), the reduced iNOS expression did not appear to be due to reductions in IFN-γ production from the T cell compartment within the brain, which was unchanged between groups (Supplementary Fig. 2a, b). Together, these data suggest that the CNS immune response is affected in IL-1R1 KO mice, with striking deficits, particularly in the myeloid response.

Importantly, these differences were restricted to the site of infection, as there were no deficits in any immune cell compartments in the spleens of IL-1R1 KO mice (Supplementary

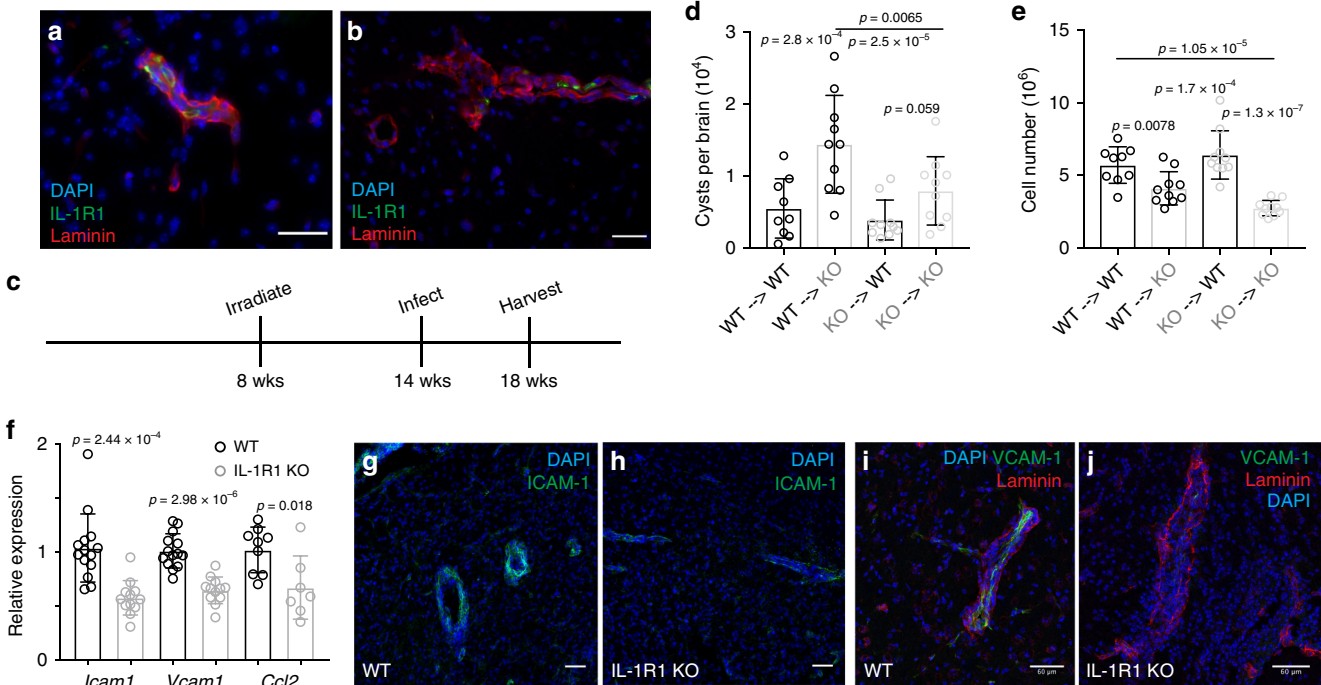

**Fig. 3 IL-1R1 is expressed by brain vasculature. a, b** Brains from chronically infected C57B6/J WT mice were sectioned and stained with DAPI (blue) and antibodies against laminin (red) and IL-1R1 (green), showing parenchymal blood vessels. **c–e** WT (CD45.1) and IL-1R1 KO (CD45.2) mice were lethally irradiated and then reconstituted with bone marrow from either WT or IL-1R1 KO mice. Mice were allowed to reconstitute for 6 weeks and then were infected i.p. with 10 cysts of the Me49 strain of *T. gondii*. 4 weeks p.i. mice were sacrificed and their brains were harvested for analysis. (*n* = 39 mice). **d** Brains were homogenized and cysts were counted by light microscopy. **e** Brains were processed for flow cytometry and the numbers of total leukocytes were calculated. Cells were pre-gated on singlets/live. **d, e** Data compiled from two experiments, statistics performed using a randomized block ANOVA (two way). Data presented as mean values ± SEM. **f** WT and IL-1R1 KO mice were infected i.p. with 10 cysts of the Me49 strain of *T. gondii*. 6 weeks p.i. the mice were sacrificed and brains were homogenized, RNA was extracted, and qPCR analysis was performed. Data compiled from 2 (*Ccl2*) or 3 (*Icam1*, *Vcam1*) experiments; statistics performed using a randomized block ANOVA (two way). Data presented as mean values ± SEM. (*n* = 26 mice for *Icam1* and *Vcam1*, *n* = 16 mice for *Ccl2*). **g–j** Brains from chronically infected WT and IL-1R1 KO mice were sectioned and stained for either ICAM-1 (**g, h**) or VCAM-1 (**i, j**). Representative images of blood vessels are shown. **g, h** Scale bars are 50 μm and **i, j** scale bars are 60 μm.

Fig. 2c–h). In fact, T cell and macrophage responses were slightly elevated in the spleen. The immune deficits in IL-1R1 KO mice are also specific to chronic infection as IL-1R1 KO mice analyzed earlier during infection (12 dpi) displayed no deficit in their monocyte/macrophage or T cell populations compared to WT in the peritoneal cavity or the spleen (Supplementary Fig. 3a, b). IFN-γ levels in the serum were, if anything, increased in IL-1R1 KO mice at this time, indicating that this response is not impaired (Supplementary Fig. 3c). The only immune defect detected during this early phase of infection in IL-1R1 KO mice was a decrease in neutrophils recruited to the peritoneal cavity (Supplementary Fig. 3a). In sum, these results show that mice lacking IL-1R1 have an impaired response of blood-derived immune cells in the brain, leading to increased parasite burden. This suggests that IL-1 signaling promotes immune responses in the brain during chronic *T. gondii* infection.

**IL-1R1 is expressed by brain vasculature.** Having established a role for IL-1 signaling in promoting the immune response to chronic *T. gondii* infection in the brain, we wanted to determine which cells in the brain can respond to IL-1 in the brain environment. We performed immunohistochemical staining for IL-1R1 on brain sections from chronically infected mice. We found that IL-1R1 was expressed principally on blood vessels in the brain, as marked by laminin staining which highlights basement membranes of blood vessels (Fig. 3a, b). Interestingly, expression is not seen continuously along vessels (Fig. 3a, b), nor on all vessels (Fig. 3b). This suggests a degree of heterogeneity among

endothelial cells and perhaps in their ability to respond to IL-1. We detected IL-1R1 expression specifically on CD31+ cells by IHC (Supplementary Fig. 4a) and by flow cytometry (Supplementary Fig. 4b, c). To test whether endothelial expression of IL-1R1 is required in this infection, we first assessed potential contributions from radiosensitive (hematopoietic) and radio-resistant (non-hematopoietic) cells. To do this, we created bone marrow chimeras with IL-1R1 KO mice. We lethally irradiated both WT and IL-1R1 KO mice, and then i.v. transferred bone marrow cells from either WT or IL-1R1 KO mice. We allowed 6 weeks for reconstitution before infecting with *T. gondii*, and we performed our analyses at 4 weeks post infection (Fig. 3c). We found that IL-1R1 KO recipients that had received WT bone marrow, had a higher brain cyst burden than WT recipients that had received either WT or IL-1R1 KO bone marrow (Fig. 3d). Consistent with this, IL-1R1 KO recipient mice, regardless of their source of bone marrow displayed a decrease in total leukocyte numbers in the brain compared to WT recipients (Fig. 3e). Taken together, these data suggest that IL-1R1 expression on a radio-resistant cell population is required for host control of the parasite, which is consistent with our hypothesis that the relevant expression is on brain endothelial cells.

**Adhesion molecule expression is dependent on IL-1R1.** During chronic *T. gondii* infection continual infiltration of immune cells into the brain is necessary for maintaining control of the parasite. One step in getting cells to successfully infiltrate the brain, as in other tissues, is the interaction with activated endothelium

expressing vascular adhesion molecules as well as chemokines. Indeed, the brain endothelium is activated during chronic *T. gondii* infection compared to the naïve state, as seen by increased expression of ICAM-1 and VCAM-1 molecules on brain endothelial cells (Supplementary Fig. 5a–d). Our data show that ICAM-1 is expressed to a higher extent by endothelial cells that express IL-1R1 compared to cells that do not in the naïve state (Supplementary Fig. 5e), and that IL-1R1$^+$ endothelial cells also express VCAM-1 in infected tissues (Supplementary Fig. 5f).

We investigated the dependence of these molecules on IL-1 signaling in our model, and found that their expression is dependent in part on IL-1 signaling. IL-1R1 KO mice displayed decreased mRNA expression of *Icam1*, *Vcam1*, and *Ccl2* in the brain (Fig. 3f) as assessed using whole-brain homogenate from chronically infected mice. To more specifically address effects on the CNS vasculature, we examined expression of ICAM-1 and VCAM-1 protein in brain sections of WT and IL-1R1 KO mice during chronic infection using IHC (Fig. 3g–j). Representative images show a marked decrease in ICAM-1 and VCAM-1 reactivity on blood vessels in the brains of IL-1R1 KO mice compared to WT (Fig. 3g–j). Together, these data show that the increased expression of vascular adhesion molecules, and potentially chemokine, in the brain that is characteristic of chronic *T. gondii* infection is partially dependent on IL-1 signaling. The modulation of adhesion molecule expression may be one mechanism by which IL-1 promotes the infiltration of immune cells into the brain during chronic *T. gondii* infection.

To determine the importance of ICAM-1 and VCAM-1 in the recruitment of infiltrating monocytes during chronic *T. gondii* infection, we used antibody treatments to block their ligands (LFA-1 and VLA-4, respectively) in vivo. We treated chronically infected WT mice with a combination of α-LFA-1 and α-VLA-4 blocking antibodies, giving a total of two treatments. After 5 days of treatment, mice receiving blocking antibody displayed decreases in the number of infiltrating myeloid cells isolated from the brain compared to control-treated mice (Supplementary Fig. 5g). Specifically, we observed deficits in the Ly6C$^{hi}$ population (Supplementary Fig. 5h), indicating a lack of blood-derived monocytes. The decrease in monocyte entry translated into fewer iNOS$^+$ cells in the brain (Supplementary Fig. 5i). These data show that interactions with ICAM-1 and VCAM-1 are necessary for monocyte infiltration into the brain during chronic infection, and that IL-1 signaling promotes the expression of these adhesion molecules.

**IL-1α KO mice have an impaired immune response to infection**. IL-1α and IL-1β both bind to and signal through IL-1R1. Having established a role for IL-1 signaling in promoting the myeloid response in the brain during chronic *T. gondii* infection, we sought to determine whether this effect was mediated by one or both of these cytokines, given that IL-1α and IL-1β are expressed by different populations of myeloid cells in the infected brain. To address this, we infected mice lacking either IL-1α or IL-1β and analyzed the cellular immune response and parasite burden during chronic phase of infection. At six weeks post-infection, IL-1α KO mice displayed an increase in parasite burden compared to WT as measured by qPCR analysis of parasite DNA from brain homogenate (Fig. 4a). IL-1β KO mice, however, showed no change in parasite burden compared to WT (Fig. 4a). This suggests that, rather unexpectedly, IL-1α is involved in maintaining control of the parasite during chronic infection, while IL-1β is not.

IL-1α KO mice displayed fewer focal areas of inflammation compared to WT (Fig. 4b, c), as seen by clusters of immune cells in H&E stained brain sections. We further found that IL-1α KO

mice, like IL-1R1 KO mice, have decreases in peripheral monocyte/macrophage populations infiltrating the brain as well as a decrease in the number of iNOS-expressing cells compared to WT (Fig. 4d, e). They also had a decrease in CD8$^+$ T cells in the brain (Fig. 4f, g). On the other hand, IL-1β KO mice displayed no difference from WT in the number of peripheral myeloid cells infiltrating the brain during chronic infection, or in the number of these cells that are expressing iNOS across multiple experiments (Fig. 4h, i), which is consistent with no change in parasite burden in these mice. IL-1β KO mice also showed no defect in T cell infiltration (Fig. 4j, k). Together, these results suggest that the role of IL-1 signaling in promoting immune responses in the brain during chronic *T. gondii* infection is mediated by IL-1α, rather than by IL-1β.

**IL-1α is released by microglia isolated from infected brains**. Our results demonstrate a role for IL-1α in chronic *T. gondii* infection. We have also shown that microglia in the infected brain are enriched in IL-1α compared to macrophages, though it is expressed by both populations. Thus, we aimed to determine which cell type releases IL-1α in this model. Uninfected mice treated with PLX5622 for 12 days to deplete microglia lost almost all *Il1a* mRNA expression in the brain (Fig. 5a), consistent with flow cytometry and immunohistochemistry data detecting IL-1α in microglia in naïve mice (Supplementary Fig. 1a, b, d). To examine IL-1α release during infection, we first established an assay to measure IL-1α release from isolated brain cells ex vivo. A single cell suspension was generated from brain homogenate, brain mononuclear cells were washed and plated in complete media for 18 hours, and the supernatant was collected for analysis by ELISA. Using this method, we found that cells isolated from mouse brain can indeed release IL-1α in an infection-dependent manner (Fig. 5b). It should be noted that splenocytes from infected animals did not release detectable IL-1α. We then used our microglia reporter model to FACS sort ZsGreen$^+$ microglia and ZsGreen$^-$ myeloid cells from infected mice. Equal numbers of microglia and peripheral myeloid cells were plated and the supernatant was collected to measure IL-1α release. We observed a very clear difference in these populations; purified microglia released IL-1α ex vivo, while purified monocytes/macrophages released negligible amounts of this cytokine (Fig. 5c). We show that this difference in IL-1α release does not appear to do due to overall increased death in microglia ex vivo as blood-derived cells actually released more LDH (Fig. 5d). We also show that IL-1α release is inhibited when membrane integrity is preserved with glycine treatment (Fig. 5e) as well as the total possible IL-1α release from isolated brain mononuclear cells ex vivo (Fig. 5f). Taken together, these findings show that microglia from infected mice have the capability to release IL-1α, which could suggest that microglia and macrophages may undergo different types of cell death.

**Caspase-1/11 KO mice have an impaired response to infection**. To begin to address whether inflammatory cell death could release IL-1α in the brain during chronic *T. gondii* infection, we first took a broad look at cell death in the brain. 4 weeks p.i., mice were injected intraperitoneally (i.p.) with propidium iodide (PI). 24 h after PI injection, mice were sacrificed for analysis. PI uptake in cells, which is indicative of cell death or severe membrane damage, was observed in the brains of *T. gondii* infected mice, and appeared in focal areas (Fig. 6a, b), suggesting that there is cell death occurring in the brain during chronic infection.

Inflammasome activation has been implicated in vitro and during acute *T. gondii* infection[39–41], and could potentially be involved in IL-1α release. IL-1α, like IL-1β, is not canonically

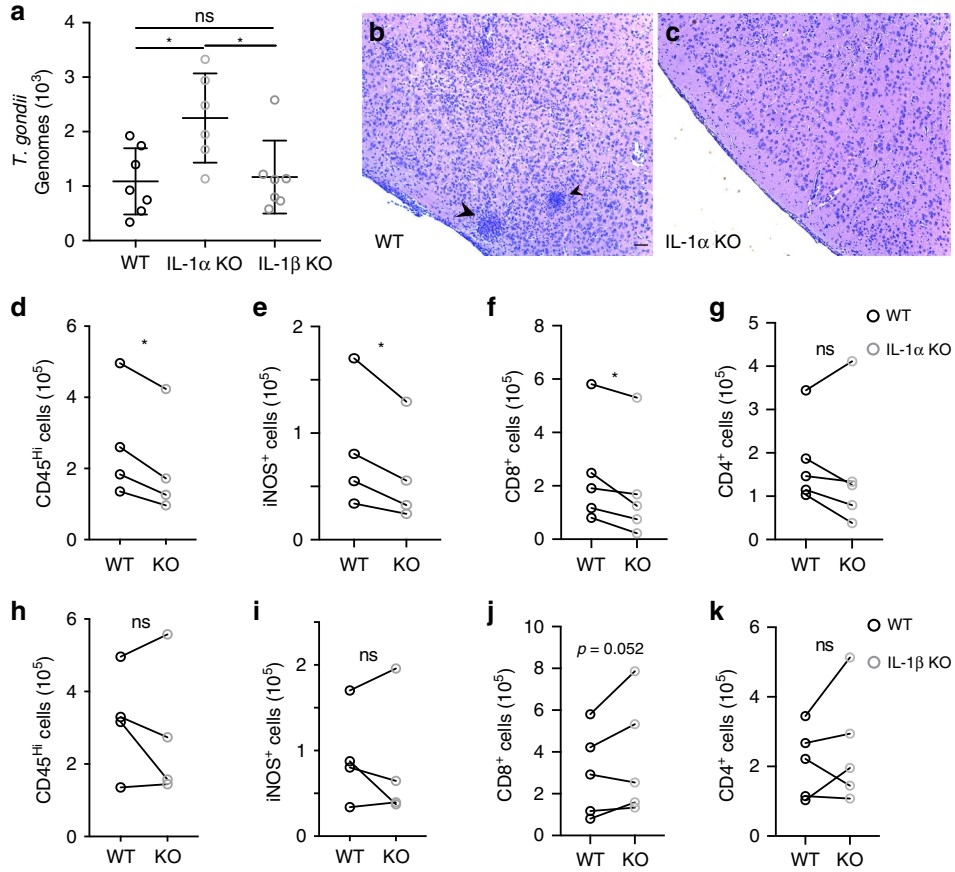

**Fig. 4 IL-1α KO mice have an impaired immune response to infection.** WT C57B6/J, IL-1α KO, and IL-1β KO were infected i.p. with 10 cysts of the Me49 strain of T. gondii. 6 weeks p.i. brains were harvested and analyzed. **a** Genomic DNA was isolated from brain homogenate, and parasite DNA was quantified using real-time qPCR. Data compiled from two experiments; statistics performed using a randomized block ANOVA (two way). Data presented as mean values ± SEM. (n = 20 mice) For WT vs IL-1α KO p = 0.0259, for IL-1α KO vs IL-1β KO p = 0.0198. **b, c** Brain slices from WT (**b**) and IL-1α KO (**c**) were H&E stained and representative images are shown. Arrow heads indicate clusters of immune cells. Scale bar indicates 100 μm (images were taken at the same magnification). **d–k** Brains were processed to obtain a single cell suspension, and analyzed by flow cytometry. Paired averages from 4 or 5 compiled experiments, statistics performed using a randomized block ANOVA (two way). **d, h** Blood-derived myeloid cells per brain as defined by CD11b+CD45hi. Cells were pre-gated on singlets/live/CD45+/CD11c−. (n = 32 mice). **d** p = 0.039. **e, i** iNOS+ cells per brain were quantified, pre-gated on singlets/live/CD45+/CD11c−/CD11b+CD45hi. (n = 32 mice). **e** p = 0.045. **f, g, j-k** CD8+ and CD4+ T cells were quantified, pre-gated on singlets/live/CD3+. **f** p = 0.033. (**f, g,** n = 37 mice; **j, k,** n = 40 mice). Source data (**d–k**) are provided as a Source data file.

secreted and requires cell death or significant membrane perturbation to be released extracellularly[46–48]. IL-1α does not need to be processed by the inflammasome platform, however, because permeabilization of the plasma membrane is required for IL-1α to be released, inflammasome-mediated cell death may still contribute to its release. To look for evidence of inflammasome activation in the brains of mice chronically infected with *T. gondii*, we infected ASC-citrine reporter mice, in which the inflammasome adaptor protein apoptosis-associated speck-like protein containing CARD (ASC) is fused with the fluorescent protein citrine. Upon inflammasome activation, the reporter shows speck-like aggregates of tagged ASC. In the brain during chronic *T. gondii* infection, ASC specks were observed around areas of inflammation in Iba1+ microglia or macrophages (Fig. 6c). We further crossed the ASC-citrine mouse line to the microglia reporter mouse line. Following infection, ASC specks were observed contained within microglia in the infected brain (Fig. 6d).

To further investigate a role for inflammasome-dependent processes in chronic *T. gondii* infection, we infected caspase-1/11 KO mice. Six weeks p.i., mice lacking these inflammatory caspases had an increased number of parasite cysts in their brains (Fig. 6e), indicating impaired parasite control. Caspase-

1/11 KO mice also have a decrease in the number of cells of the monocyte/macrophage lineage in the brain during chronic infection (Fig. 6f), as well as significantly fewer infiltrating myeloid cells expressing iNOS compared to WT mice (Fig. 6g). These mice also displayed decreases in CD4+ T cells (Fig. 6h, i). In addition to an increased overall cyst burden, caspase-1/11 KO mice had more instances of clusters of parasite cysts compared to WT (Fig. 6j, k), likely indicating a lack of parasite control in areas of parasite reactivation. Together, these results are similar to those observed in infected IL-1R1 KO mice and show that caspase-1/11 activity is important for host control of *T. gondii* infection.

**Inflammation and IL-1α release depend on gasdermin-D**. Our data implicate an inflammasome-dependent processes in the control of *T. gondii* in the brain, thus we investigated the importance of gasdermin-D, the pore-forming executor of pyroptosis[37,49–51]. We utilized gasdermin-D (gsdmd) KO mice to specifically assess the importance of pyroptosis. Six weeks p.i., gsdmd KO mice displayed a significant increase in parasite cyst burden compared to WT (Fig. 7a). Like IL-1R1 KO, IL-1α KO, and caspase-1/11 KO mice, gsdmd KO mice also displayed a

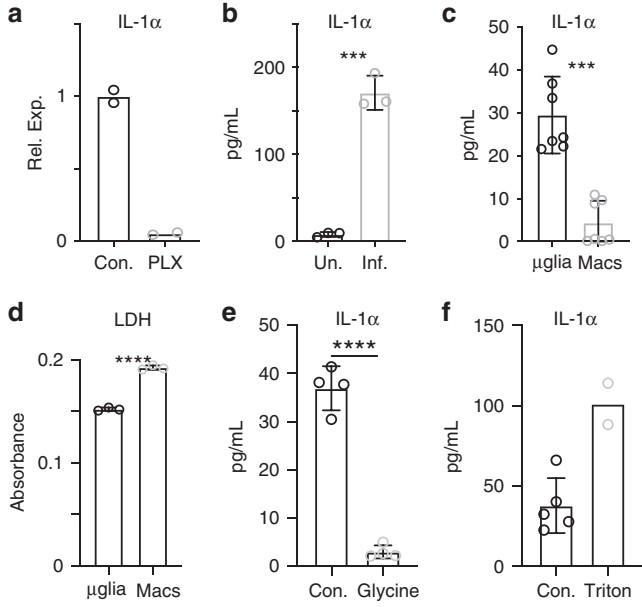

**Fig. 5 IL-1α is released by microglia isolated from infected brains.** Uninfected mice were fed either control chow or chow containing PLX5622 for 12 days prior to sacrifice. **a** mRNA levels of IL-1α were determined by RT-qPCR on whole-brain homogenate. ($n = 2$ mice per group) Data are presented as mean values. **b** 6 weeks p.i. brains from WT mice were harvested and processed to a single cell suspension. Cells were plated in a 96 well plate and incubated at 37 °C overnight. IL-1α release was then measured by ELISA. ($n = 3$ mice per group, $p = 0.0001$) Data are presented as mean values ± SEM. **c, d** Chronically infected $CX_3CR_1^{Cre\text{-}ERT2}$ × ZsGreen$^{fl/stop/fl}$ mice were sacrificed and brains were harvested and processed for flow cytometry. Samples were run on a BD Aria, gated on live/singlets/CD45$^+$/CD11b$^+$ from which ZsGreen$^+$ and ZsGreen$^-$ populations were gated and sorted. Cells from 6 mice were pooled. Equal numbers of each cell population were plated and incubated overnight at 37°C. Supernatants were collected and analyzed by ELISA for IL-1α (**c**, $n = 7$ wells per group, $p = 9.76 \times 10^{-9}$) and LDH (**d**, $n = 3$ wells per group, $p < 0.0001$) (plotted as absorbance at 490–680 nm). For **c** results from two experiments are shown. Data are presented as mean values ± SEM. **e, f** Assay was performed as in (**b**), with some wells treated with glycine to stop membrane permeability (**e**, $n = 4$ wells per group, $p < 0.0001$) or triton-containing lysis buffer to show total possible release (**f**, $n = 7$ wells). Data are presented as mean values ± SEM. Statistics were performed using a two-tailed Student's T test (**a, b, d, e**) or a Randomized Block ANOVA (two way) (**c**).

decrease in the number of immune cells infiltrating the brain (Fig. 7b).

To directly assess the contribution of pyroptosis to IL-1α release, brain mononuclear cells were isolated from gsdmd KO mice and ex vivo IL-1α release was determined by ELISA. Cells isolated from the brains of gsdmd KO mice released significantly less IL-1α into the supernatant than cells from WT mice, about a 70 percent reduction in IL-1α release (Fig. 7c). We also utilized necrosulfonamide (NSA), which has been shown to be a specific inhibitor of gsdmd in mice[52]. Brain cells isolated from infected WT mice were analyzed for ex vivo IL-1α release under control conditions, or incubated with 20 μM NSA (Fig. 7d). Strikingly, NSA inhibited ex vivo IL-1α release, confirming that release is dependent on gsdmd. Taken together, these results suggest that IL-1α is released from cells from infected brains in a gsdmd-dependent manner, and promotes the infiltration of anti-parasitic immune cells into the *T. gondii* infected brain.

## Discussion

*Toxoplasma gondii* establishes a chronic brain infection in its host, necessitating long-term neuroinflammation[4,5,53]. Much is known about the immune response to this parasite, but the functions of the brain resident microglia in the chronic stage of infection are still being understood. Early studies using culture systems of murine and human microglia showed that IFN-γ and LPS treatment prior to infection inhibited parasite replication[27–29]. However, understanding of microglia-specific functions in brain infections has been hindered by the fact that microglia rapidly lose their identity in culture[54]. Moreover, culture techniques do not recapitulate the complex interactions microglia have during infection with other cells or the tissue architecture of the brain. Studies in mice have shown that microglia and infiltrating macrophages can produce chemokines and cytokines in infection[9,14,30–32]. However, our microglia reporter mouse model has allowed us to uncover functions specific to microglia, distinct from that of infiltrating macrophages, during chronic infection with *T. gondii*. Thus, we aimed to examine microglia and macrophages within the brain to investigate potential differences.

Through RNA-seq analysis as well as staining of infected brain tissue, we find that there is an NF-κB signature present in brain-infiltrating monocytes/macrophages, that is largely absent in microglia in the same environment. These two cell types are likely exposed to the same signals within the brain, which suggests that the ontogeny of these cells has long-lasting implications for their functional capacity. Transcription factors, including *Sall1*, that define microglia identity may be shaping the transcriptional landscape, repressing loci that could potentially lead to damaging inflammation in the brain[55]. We suggest that these differences are evidence of a division of labor, with microglia and blood-derived macrophages contributing in different ways to inflammation in the brain during infection with *T. gondii*. While blood-derived cells display a classic inflammation-associated NF-κB response, microglia may be better suited to contributing to inflammation through the release of alarmins, rather than through upregulation of a broader program that may be injurious to the local tissue.

We find the alarmin IL-1α expressed in microglia, though they notably lack expression of IL-1β which is found in infiltrating myeloid cells. This suggests that both of these cell types may be able to participate in an IL-1 response, but in fundamentally different ways. Importantly, we show that host immunity is dependent on the activity of IL-1α rather than IL-1β, and that IL-1α is released ex vivo from microglia but not from infiltrating macrophages. In general, IL-1β has been the subject of more study than IL-1α, and has a history of being implicated when IL-1 signaling is discussed. More recently, IL-1α has been shown to contribute to certain inflammatory environments. IL-1α has been shown to initiate sterile lung inflammation in response to silica[56]. Recently, it was suggested that IL-1α rather than IL-1β drives sepsis pathology[57]. IL-1α activity in the CNS has begun to be studied, with a deleterious role for the cytokine shown in spinal cord injury[58]. IL-1β has been implicated in some infection models, but IL-1α activity in brain infection has not previously been reported. As an alarmin expressed in the brain at baseline, IL-1α is ideally placed to initiate inflammation in response to early damage caused by the parasite before there is robust immune infiltration.

In this work, we also show that IL-1α likely signals on brain vasculature, promoting the infiltration of immune cells. We found that IL-1R1 expression on brain vasculature displays a mosaic pattern. This could suggest that there are functionally distinct sub-populations of endothelial cells capable of becoming activated in response to different signals[59]. There is ample evidence in the literature to support IL-1R1 expression on endothelial cells as well as the responsiveness of CNS

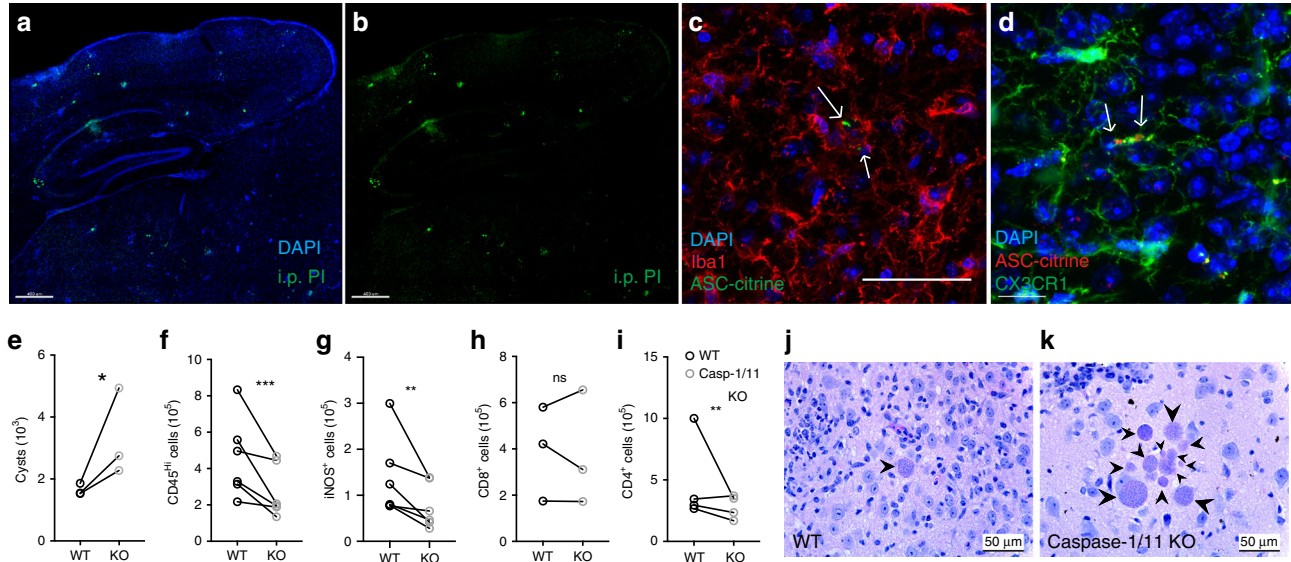

**Fig. 6 Caspase-1/11 KO mice have an impaired response to infection. a, b** Chronically infected C57B6/J mice were injected i.p. with 20 mg/kg propidium iodide. 24 h later, mice were sacrificed and brains were imaged with confocal microscopy. A representative image is shown. **c, d** Mice expressing ASC-citrine (**c**) or ASC-citrine and CX3CR1-cre$^{ERT2}$ZsGreen (**d**) were infected with 10 cysts of the Me49 strain of *T. gondii*. 4 weeks post infection brains were harvested, cryopreserved, stained, and imaged. Arrows indicate ASC aggregates in Iba1$^+$ cells (**c**) or in ZsGreen$^+$ microglial cells (**d**). **e-i** WT and casp-1/11KO mice were infected with 10 cysts of the Me49 strain of *T. gondii*. 6 weeks p.i. brains were harvested and analyzed. Paired averages for 3–6 experiments are shown. **e** Cyst burden per brain was determined by counting cysts in brain homogenate on a light microscope. ($n = 20$ mice, $p = 0.034$). **f** Infiltrating myeloid cell populations were quantified by flow cytometry. Cells were pre-gated on singlets/live/CD45$^+$/CD11c$^-$. ($n = 51$ mice, $p = 1.47 \times 10^{-4}$). **g** iNOS$^+$ cell populations were quantified, cells were pre-gated on singlets/live/CD45$^+$/CD11c$^-$/CD11b$^+$/CD45hi. ($n = 51$ mice, $p = 0.0024$). **h, i** CD8$^+$ and CD4$^+$ T cell populations were quantified, cells were pre-gated on live/singlets/CD3$^+$. **h** ($n = 41$ mice). **g** ($n = 51$ mice, $p = 7.47 \times 10^{-4}$). **j, k** Brain slices from WT (**j**) and caspase-1/11 KO (**k**) mice were H&E stained and representative images are shown. Arrow heads indicate parasite cysts. Statistics were performed using a randomized block ANOVA (two way) (**e-i**). Scale bars in (**a, b**) are 400 μm, scale bar in (**d**) is 15 μm, all other scale bars are 50 μm. Source data (**e-i**) are provided as a Sourcedata file.

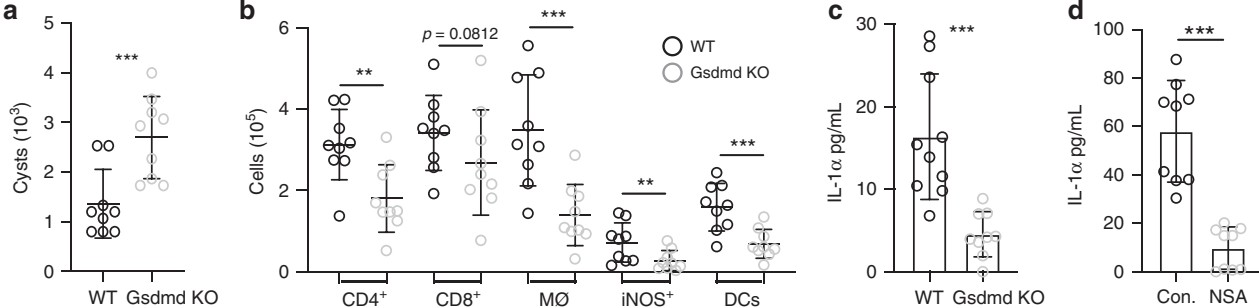

**Fig. 7 Inflammation and IL-1α release depend on gasdermin-D. a–c** C57B6/J and Gasdermin D KO mice were infected i.p. with 10 cysts of the Me49 strain of *T. gondii*. 6 weeks p.i., mice were sacrificed and tissues were harvested for analysis. Data from two experiments are shown ($n = 18$ mice). **a** Cyst burden per brain was determined by counting cysts in brain homogenate on a light microscope, $p = 8.14 \times 10^{-4}$. **b** Brain tissue was processed for flow cytometry analysis and immune cell populations were quantified. All populations were previously gated on live/singlets. CD4$^+$ and CD8$^+$ were pre-gated on CD3$^+$ T cells; DCs were pre-gated on CD45$^+$ cells; infiltrating macrophage/monocytes (Mφ) are defined as CD11c$^-$CD11b$^+$CD45hi; iNOS$^+$ cells were gated within the Mφ gate. $p$ values from left to right are: 0.0017, 0.0812, 0.00048, 0.0012, and 0.00022. **c** Single cell suspension from brain homogenate from WT and gsdmd KO mice was plated in a 96 well plate and incubated at 37 °C overnight. Supernatant was isolated and analyzed by ELISA for IL-1α, $p = 4.77 \times 10^{-4}$. **d** Brain homogenate from WT mice was plated as in (**c**), with either control media or 20 μm necrosulfonamide (NSA), $p = 8.05 \times 10^{-10}$. IL-1α release from two experiments is shown (**d**). ($n = 9$ mice) Data are presented as mean values ± SEM. Statistics were performed using a randomized block ANOVA (two way).

vasculature to IL-1[60–64]. However, IL-1 has also been shown to signal on immune cells[65–67]. We found that IL-1R1 expression on radio-resistant cells is important in our model, which is supportive of endothelial cells being the relevant responders, but there has also been evidence put forth that other brain resident cells can respond to IL-1. It has been suggested that microglial IL-1R1 expression plays a role in self-renewal after ablation[68]. Microglia are partially radio-resistant and do experience some turnover after irradiation and repopulation. It has also been suggested that IL-1 can act on neurons, though it should be noted that it has also been reported that neurons express a unique form of IL-1RAcP which affects downstream signaling[69]. It is unclear whether astrocytes express IL-1R1, but astrocytes represent another radio-resistant cell population in the brain that has the ability to affect immune cell infiltration through chemokine production[14,43].

Infiltrating immune cells express pro-IL-1β but we have not detected a role for IL-1β in this model. We have also shown that they do not release IL-1α ex vivo even though they express it, suggesting that they may die in an immunologically quiet way such as apoptosis, while microglia may undergo a more inflammatory form of cell death, including pyroptosis. If these two cell types do undergo different forms of cell death, it is of great interest how microglia activate gasdermin-D to release inflammatory factors. It is possible that microglia in an area of parasite reactivation become infected, sense parasite products in the cytoplasm, and undergo death to eliminate this niche for parasite replication. NLRP1 and NLRP3 have both been shown to recognize _T. gondii_[39–41] and could be candidate sensors in microglia. AIM2 can recognize DNA[70] and could therefore be activated if parasite DNA becomes exposed to the cytosol. However, we and others[6] have not been able to observe direct infection of microglia in chronically infected mice. Microglia migrating to sites of parasite reactivation may alternatively recognize products resulting from host cell death or damage, such as ATP, and undergo death that will promote inflammation. Thus, the sensors upstream of caspase-dependent cleavage of gasdermin-D in microglia are of great interest.

## Methods

**Mice and infections**. C57BL/6 (CD45.2) mice were purchased from The Jackson Laboratory or bred within our animal facility in specific pathogen-free facilities. B6. SJL-Ptprc$^a$ Pepc$^b$/BoyCrCl (C57BL/6 CD45.1) mice used in the bone marrow chimera experiments were purchased from Charles River. _Gsdmd_$^{-/-}$ mice were provided by J. Lukens. All other mouse lines were originally purchased from Jackson Laboratories and were then maintained within our animal facility. These include: _Il1a_$^{-/-}$ (#67031), _Il1b_$^{-/-}$ (#034447), _Il1r1_$^{-/-}$ (#003245), _Casp1_$^{-/-}$_Casp11_$^{-/-}$ (#016621), ASC citrine (#030744), and CX3CR1$^{Cre-ERT2}$ × ZsGreen$^{fl/stop/fl}$ mice. In the case of the CX3CR1$^{Cre-ERT2}$ × ZsGreen$^{fl/stop/fl}$ mice, CX3CR1$^{Cre-ERT2}$ mice (#020940) and Ai6 mice (#007906) were purchased from Jackson Laboratories and cross-bred within our facility. All mice were housed in UVA specific pathogen-free facilities with a 12 h light/dark cycle, ambient temperature between 68 and 72 °C, and humidity between 30 and 60%. All mice were age- and sex-matched for all experiments. Infections used the type II _T. gondii_ strain Me49, which was maintained in chronically infected Swiss Webster mice (Charles River Laboratories) and passaged through CBA/J mice (The Jackson Laboratory) for experimental infections. For the experimental infections, the brains of chronically infected (4–8 week) CBA/J mice were homogenized to isolate tissue cysts. Experimental mice were then injected i.p. with 10 Me49 cysts. All procedures followed the regulations of, and were approved by, the Institutional Animal Care and Use Committee at the University of Virginia.

**Sampling**. Within a single experiment, when multiple parameters were assessed in the same tissue, the same samples were used (i.e. for analysis of multiple immune cell populations by flow cytometry, cells from the same brain sample were used). Representative IHC images accompanying flow cytometry data were taken from distinct brain samples.

**_T. gondii_ cyst counts**. Brain tissue was placed in complete RPMI, minced with a razor blade, and then passed through an 18-gauge needle. 30 μL of homogenate was placed on a microscope slide and covered with a coverslip. Cysts were counted manually on a brightfield DM 2000 LED microscope (Leica Biosystems).

**Tissue processing**. Immediately after sacrifice mice were perfused with 30 mL of cold 1X PBS. Brains and spleens were harvested and put into cold complete RPMI media (cRPMI) (10% FBS, 1% penicillin/streptomycin, 1% sodium pyruvate, 1% non-essential amino acids, and 0.1% 2-ME). If peritoneal lavage fluid was collected, prior to perfusion, 5 mL of cold 1X PBS was injected through the intact peritoneal membrane with a 26-gauge needle, and removed with a 22-gauge needle. If serum was collected, blood from the heart was collected and allowed to clot at 4 °C overnight to separate serum.

After harvest, brains were minced with a razor blade, passed through an 18-gauge needle, and then enzymatically digested with 0.227 mg/mL collagenase/dispase and 50 U/mL DNase (Roche) at 37 °C for 45 min. After digestion, brains homogenate was passed through a 70 μm filter (Corning) and washed with cRPMI. To remove myelin from samples, filtered brain homogenate was then resuspended with 20 mL of 40% Percoll and spun at 650 × g for 25 min. Myelin was aspirated, samples were washed with cRPMI, and then resuspended in cRPMI. Spleens were mechanically homogenized and passed through a 40 μm filter (Corning). Samples were washed with cRPMI and then resuspended in 2 mL of RBC lysis buffer

(0.16 M NH$_4$Cl) for 2 min. Cells were then washed with cRPMI and then resuspended. Peritoneal lavage fluid was washed with cRPMI, pelleted and resuspended.

**Cytospin**. Peritoneal lavage fluid samples were diluted to $1 × 10^5$ cells/200 μL which was added to the upper chamber of the slide attachment (Simport). Samples were spun onto slides using a Cytospin 4 (Thermo Scientific), and then H&E stained.

**Flow cytometry and cell sorting**. Single-cell suspensions from tissue samples were plated in a 96-well U-bottom plate. Cells were initially incubated with 50 μL Fc block (1 μg/mL 2.4G2 Ab (BioXCell), 0.1% rat γ globulin (Jackson ImmunoResearch)) for 10 min at room temperature. Cells were then surface stained with antibodies and a Live/Dead stain for 30 min at 4 °C. After surface staining, cells were washed with FACS buffer (0.2% BSA and 2 mM EDTA in 1× PBS) and fixed at 4 °C for 30 min with either 2% paraformaldehyde (PFA) or a fixation/permeabilization kit (eBioscience). Cells were then permeabilized and stained with any intracellular markers for 30 min at 4 °C. Samples were then washed, resuspended in FACS buffer, and run on a Gallios flow cytometry (Beckman Coulter). Analysis was done using FlowJo software v.10. Antibody clones used include: CD31 (390), CD45 (30-F11), MHC-II (M5/114.15.2), NK1.1 (PK136), CD19 (1D3), CD3 (17A2), CD4 (GK1.5), CD11c (N418), CD11b (M1/70), Foxp3 (FJK-16s) 1:400, Ly6G (1A8), Ly6C (HK1.4), CD8a (53-6.7), IFN-γ (XMG1.2), NOS2 (CXNFT) 1:400, IL-1α (ALF-161), and pro-IL-1β (NJTEN3). Dilutions for all antibodies are 1:200 unless otherwise noted.

For cell sorting, CX3CR1cre$^{ERT2}$ × ZsGreen$^{fl/stop/fl}$ mice were used. After surface staining, live cells were analyzed on a BD Aria in the UVA flow cytometry core facility. Cells were sorted based on ZsGreen expression, into serum-containing media for ex vivo culture, or into Trizol for RNA-sequencing.

**Ex vivo culture experiments**. To assess IL-1 release, brain single-cell suspensions were plated in a 96-well plate in complete RPMI media with no additional stimulation. They were then incubated at 37 °C for 4 h to overnight (indicated in figure legends). For gasdermin-D inhibition assays, 20 μM necrosulfonamide (NSA) was added to wells during incubation. Cells were pelleted and supernatants were collected for analysis. For sorted cell populations, equal numbers of microglia and macrophages were plated for analysis.

**Quantitative RT-PCR**. Approximately ¼ of a mouse brain was put into 1 mL TRIzol (Ambion) in bead-beating tubes (Sarstedt) containing 1 mm zirconia/silica beads (BioSpec). Tissue was homogenized for 30 s using a Mini-bead beater (BioSpec). RNA was extracted according to the manufacturer's instructions (Ambion). High Capacity Reverse Transcription Kit (Applied Biosystems) was used for cDNA synthesis. qRT-PCR was done using 2X Taq-based Master Mix (Bioline) and TaqMan gene expression assays (Applied Biosystems). Reactions were run on a CFX384 Real-Time System (Bio-Rad Laboratories). HPRT was used as the housekeeping gene for all reactions and relative expression was calculated as $2^{(-\Delta\Delta CT)}$. Primer assays were acquired from ThermoFisher and include Vcam1 (Mm01320970_m1), Icam1 (Mm00516023_m1), Ccl2 (Mm00441242_m1), and Il1a (Mm00439620_m1).

**Immunohistochemistry**. Brains from mice were harvested and placed in 4% PFA for 24 h. Following PFA fixation, brains were moved to a solution of 30% sucrose for 24 h, and were then embedded in OCT and flash frozen on dry ice. Samples were stored at −20 °C until cutting. After cutting, sections were blocked in 1× PBS containing 0.1% triton, 0.05% Tween 20, and 2% goat or donkey serum (Jackson ImmunoResearch) for 1 h at room temperature. Sections were then incubated with primary Abs overnight at 4 °C. Sections were then washed with PBS, and incubated with secondary Abs for 1 h at room temperature. Sections were then washed, and nuclear stained with DAPI (Thermo Fisher Scientific) for 5 min at room temperature. Finally, sections were mounted, covered in Aquamount (Lerner Laboratories), and covered with coverslips (Thermo Fisher Scientific). All images were captured using a Leica TCS SP8 Confocal microscope system. Images were analyzed using either ImageJ or Imaris software.

Antibodies used include: pro-IL-1β (NJTEN3), iNOS (product # PA5-16855) 1:400, CD31 (390), VCAM-1 (429), ICAM-1 (YN1/1.7.4), c−Rel (G-7), IL-1α (ALF-161) 1:100, IL-1R1 (cat # AF771) 1:100, p65 (D14E12), Iba1 (cat # 019-19741) 1:1000, and laminin (product # CL54851AP-1). Dilutions are 1:200 unless otherwise noted.

**H&E Tissue sections**. Brains from mice were submerged in formalin and sent to the UVA Research Histology Core, where they were embedded in paraffin, sectioned and stained with hematoxylin and eosin. They were then imaged on a brightfield DM 2000 LED microscope (Leica Biosystems).

**ELISAs**. Samples for ELISAs were obtained by harvesting mouse brains and processing them to form a single cell suspension. Cells were then plated in 96-well plates and incubated at 37 °C either overnight or for 5 h (indicated in figure

legends). Supernatants were then collected and stored at −20 °C until use. ELISAs for IL-1α (BioLegend 433401) and IL-1β (BioLegend 432601), as well as for IFN-γ, were performed according to the manufacturer's instructions. Briefly, Immunolon 4HBX ELISA plates (Thermo Fisher Scientific) were coated with capture antibody at 4 °C overnight. Plates were then washed and blocked with buffer containing BSA at room temperature for 1 hour. After washing, standards and samples were added and incubated at room temperature for 2 h. After washing, biotinylated detection antibody was added and incubated for 1 h at room temperature. Plates were washed and incubated with avidin-HRP for 30 min at room temperature. Finally, plates were washed and incubated with ABTS peroxide substrate solution (SouthernBiotech) for 15 min or until color change occurred. Immediately after the color change, plates were read on an Epoch Biotek plate reader using Gen5 2.00 software.

**Antibody blockade experiments**. Chronically infected mice (4 weeks p.i.) were treated on days 1 and 3 of the treatment regimen with 200 μg i.p. each of anti-LFA-1 (Bio X Cell, clone M17/4, cat# BE0006) and anti-VLA-4 (Bio X Cell, clone PS/2, cat# BE0071) blocking antibodies or control IgG. They were then sacrificed and analyzed on day 5.

**Propidium iodide injection**. Chronically infected mice (4 weeks p.i.) were injected i.p. with 0.4 mg of propidium iodide. 24 h after injection, mice were sacrificed and their brains were PFA fixed and analyzed by confocal microscopy.

**Microglia depletion**. For studies involving microglia depletion, adult mice were fed either control chow or chow containing PLX5622 ad libitum for 12 days. At this time brains were harvested, RNA was extracted as described above, and IL-1α expression was measured by qPCR.

**Bone marrow chimeras**. At 8 weeks of age, C57B6/J and IL-1R1 KO mice were irradiated with 1000 rad. Bone marrow cells isolated from WT and IL-1R1 KO mice were then i.v. transferred (by retro-orbital injection) into the irradiated recipient mice. Mice were allowed to recover for 6 weeks after irradiation and reconstitution and were then infected. 4 weeks post infection, mice were sacrificed and tissues were harvested for analysis.

**RNA Sequencing data analysis**. Read quality profiles for raw FASTQ files was performed with FastQC (v0.11.5) before and after trimming and filtering. Read filtering and trimming was accomplished with Trimmomatic (v0.39) paired-end set to phred33 quality scoring. Reads were trimmed according to a four-base sliding window with a minimum quality score of 15 and minimum leading and trialing quality scores of 3. The minimum fragment length was set to 36. Trimmed and filtered reads were mapped to the GENCODE M13 genome and transcript abundances were quantified using Salmon (v0.8.2). Quantified transcript abundances were imported into the R programming environment and converted into ENSEMBL gene abundances with Tximport (v1.4.0). All pre-processing steps were performed within the Pypiper framework (v0.6.0) with Python version 2.7.14.

Differential expression testing was performed using the R Bioconductor package DESeq2 (v1.16.1) at a preset alpha value of 0.05. Log2 fold change values were shrunken using a normal prior distribution. Any results that lacked the replicates or had low counts were thrown out of the dataset prior to differential expression testing. Results of differential expression testing were visualized using the R package EnhancedVolcano (v1.2.0) to display transformed p-values (−log10) against the corresponding log2 fold change values. All labeled genes were manually selected from significantly differentially expressed genes in the DESeq2 results list.

Differential expression testing results were labeled as "upregulated" or downregulated" for a given pairwise comparison. All genes with a log2 fold change value above 0 and a BH adjusted p-value below 0.05 were designated upregulated and all genes with a log2 fold change value below 0 and a BH adjusted *p*-value below 0.05 were designated downregulated. Gene names for the differential expression results tables were converted from mouse ENSEMBL codeas to gene symbols with AnnotationDbi (v1.46.0). In order to determine the functional profile of the gene lists, the R package clusterProfiler (v3.12.0) was used to apply Fisher's exact test with respect to over-representation of GO terms for biological processes at all levels of the Gene Ontology Consortium hierarchy. The lists were tested against a background distribution that consisted of all genes that returned a p-value in differential expression testing. Significant GO terms had a BH adjusted *p*-value below 0.05.

GO terms were manually selected from the results output in the clusterProfiler package for plotting with the pheatmap package (v1.0.12). Each GO term-specific heatmap displays rlog-transformed abundance values that have been Z-score normalized with respect to each gene. The genes displayed were selected from clusterProfiler results for enrichment of GO terms for biological processes. Significantly enriched GO terms were also selected and plotted using the clusterProfiler dotplot function.

**Statistics**. Statistical analysis comparing two groups at a single time point was performed in Prism software using an unpaired Student's T test. When data from multiple experiments were combined, to show natural biological variability between infections, a randomized block ANOVA was performed using R v.3.4.4 software. This test was designed to assess the effect of the experimental group while controlling for any effect of experimental date, by modeling the group as a fixed effect and date as a random effect. Tests used for each figure is shown in the figure legend. All data were graphed using Prism software. Distributions were assumed to be normal. All graphs show the mean of the data, or the mean along with individual values. Error bars indicate standard deviation.

**Reporting summary**. Further information on research design is available in the Nature Research Reporting Summary linked to this article.

## Data availability
RNA sequencing data have been deposited in GEO under the accession code GSE146680. Source data are provided with this paper.

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

## Acknowledgements

The authors would like to acknowledge members of the Harris lab and center for Brain Immunology and Glia (BIG) for their input during the development of this work. We thank Marieke K. Jones for her help with statistical analysis and coding. We thank Sarah Ewald and her lab at UVA for sharing the ASC-citrine mice used in this study. We would like to acknowledge the support we received from core facilities at the University of Virginia, including the Flow Cytometry Core and the Research Histology Core. This work was funded by National Institutes of Health grants R01NS091067, R56NS106028, and R01NS112516 to T.H.H., R01NS106383 to J.R.L., T32AI007046 to S.J.B., T32GM008328 to K.M.S., and T32AI007496 to C.A.O., as well as the Carter Immunology Center Collaborative Research Grant to T.H.H., The Alzheimer's Association grant AARG-18-566113 to J.R.L., The Owens Family Foundation to J.R.L., and The University of Virginia R&D Award to J.R.L.

## Author contributions

S.J.B. designed and performed experiments, analyzed data, and wrote the paper. K.M.S. and C.A.O. helped with experiments and discussed results and implications. J.A.T. performed the RNA-seq experiment. D.J. conducted bioinformatics analyses. J.R.L. provided reagents, mice, and conceptual advice. T.H.H. supervised the project and edited the paper.

## Competing interests

The authors declare no competing interests.
