## [Peer Review File · Nature Communications]

Reviewers' comments:

Reviewer #1 (T.gondii neuro immunology)(Remarks to the Author):

Overview of Manuscript: The manuscript addresses the specific roles of microglia, in infections in the CNS, using the neurotrophic pathogen, *Toxoplasma gondii*, which establishes a chronic infection in the brain, as an experimental model. To identify the specific roles of microglia, the authors used microglia reporter mice, sorted microglia from infiltrating myeloid cells, and analyzed microglia and macrophages using RNA sequencing. Using this approach it was found that NF- κ B and inflammatory cytokines were overly expressed in macrophages vs. microglia and that IL-1 α was enriched in microglia while IL-1 β was enriched in macrophages. To probe this initial finding further the authors used a combination of knockout mice and chemical inhibition. Major findings were: 1) mice lacking IL-1R and IL-1 α but not IL-1 β , have impaired parasite control and immune cell infiltration, 2) microglia release IL-1 α via a gasdermin-D-dependent mechanism and 3) gasdermin-D and caspase 11 deficient mice show deficits in parasite control and immune cell infiltration. The authors conclude macrophages and microglia are differentially equipped to propagate inflammation and that in chronic *T. gondii* infection, microglia release IL-1 α which promotes neuroinflammation and parasite control.

The manuscript aim to define the specific role of microglia is of importance as the role of this resident immune cell in the CNS is not well understood. The main findings of this study that microglia and macrophages are different in their abilities to propagate inflammation in the brain and that microglia release of IL-1 α contributes to control of the chronic *T. gondii* infection, contributes both to the understanding of immune responses to infections in the CNS in general as well as to the understanding of the intracerebral immune response against Toxoplasmosis and will be of interest to the research community in these fields. This is a well-conducted study using cleverly designed experimental approaches with appropriate experimental controls. It is generally well written with the exception of a few issues regarding referencing and an over generalization or lack of understanding of literature and previous work on immune responses to *T. gondii* in the CNS, as detailed below.

Specific Comments:

Introduction

p. 3, 3rd line – the authors state 'As the only resident immune cell, microglia'. However, astrocytes can also serve as a resident immune effector cell in the brain....' Furthermore, as studies show astrocytes are an important resident immune effector cell controlling *T. gondii* in the brain, some mention of astrocytes in this first paragraph should be given for proper and complete context (see works by Hunter 2016, 2004; Drögemüller et al., 2008, and others).

p. 3, 2nd paragraph, 3rd sentence: The authors state in reference to the immune response to *T. gondii* "T-cell derived IFN- γ is one essential element" and cite 2 references. The literature on the immune response to *T. gondii* is voluminous and this simple statement referring to IFN- γ , as 'one' essential element is misleading about the role of IFN- γ . IFN- γ is 'the' essential cytokine controlling *T. gondii* in the brain with its role well defined by work by Y. Suzuki and numerous others. Additionally there is a wealth of knowledge of the immune response to this parasite in the brain. The work on the importance of IFN- γ in control of *T. gondii* by Suzuki should be cited as well as a more complete summary of the large body of knowledge known of the intracerebral immune response to *T. gondii*.

Again, later in this paragraph the authors state the role of microglia in the chronic infection has not been fully 'elucidated' which is true, but Suzuki has done significant work on microglia in the acute phase (2007, 2005) which should be cited and referenced for broader context of what is known about the role of microglia in Toxoplasmosis in the brain. As the work presented in this manuscript primarily addresses the role of microglia in the chronic phase of the immune response

to *T. gondii* and that aspect could be stated in the above statement to discriminate from what is known of microglia in the acute infection.

Results

p. 5, 1st paragraph – The statement “Blood-derived monocytes have been demonstrated to be important for host survival during infection, but whether microglia perform similar functions is still unknown” seems to be an overstatement and in part wrong. For example Suzuki’s work on microglia has shown that microglia are important as sentinel cells, able to detect early cerebral *Toxoplasma* proliferation and activate their IFN- γ production thus facilitating protection immunity in controlling parasite proliferation in the brain. This is essentially the same issue as noted above in the Introduction.

Discussion

p. 21 – 1st paragraph – The statement ‘Much is known about the immune response to this parasite but the role of the brain-resident microglia is still largely unknown.’ again reads as a little untrue or at least too broadly stated as the role of microglia in IFN- γ mediated protection as mentioned above is well established, and more recent evidence indicating migratory properties of infected microglia possibly serve to disseminate the parasite in the brain, are also in the literature. Hence this statement is just too general - a qualification of ‘in the chronic infection’ or some type of delineating/restricting condition, could be used to help clarify what is new in this manuscript while also indicating what is known of the understanding of microglia in *Toxoplasmosis*.

Later in this paragraph the authors discuss the *in vitro* experiments done with *T. gondii* infected microglia and discuss how *in vitro* systems don’t adequately reflect the biology/phenotype of microglia *in vivo*. These *in vitro* studies are quite old and since that time most studies addressing the role of microglia have been done *in vivo* so this statement seems to ignore or at least erroneously suggest the *in vitro* studies are the main source of information on microglia and *Toxoplasma*.

As a concluding statement to the authors, I wish to express that despite the above comments, I believe this is a well-done study is well-done that addresses an important question about the differential roles of macrophages and microglia in the brain in *T. gondii* which hasn’t received attention before. However the seeming lack of knowledge and/or understanding of the literature on work done on the role of microglia in the cerebral immune response to *T. gondii* bothered me throughout the manuscript. Nearly, all the specific comments refer to this issue, which occurred in several sections of the manuscript. However, addressing this issue I believe, will made this a better paper but also will put a greater emphasis on that in the manuscript that is truly new and noteworthy, and thus significantly improve the manuscript.

Reviewer #2 (*T.gondii* Immune responses)(Remarks to the Author):

Batisda et al follow a logical approach to delineate a molecular function for microglia in control of chronic *T. gondii* infection in the brain. They discover that rather than the expected NF κ B transcriptional signature present in infiltrating macrophages, brain microglia are skewed to express IL-1a. These findings are corroborated on protein level. Mice lacking IL-1R1 and IL-1a, but not IL-1b, exhibit impaired parasite control and immune cell infiltration into the brain in the chronic phase. Microglia *ex vivo* are shown to produce IL-1a, dependent on gasdermin D. Gasdermin D and caspase 1/11 knockout mice equally present with decreased parasite control and defect immune cell infiltration into the brain.

This study addresses important differences between macrophages and microglia in controlling chronic *T. gondii* by profiling their inflammatory potential in the brain. IL-1a is the “neglected” IL-1 in the context of inflammasome activation and IL-1R signalling, and this study adds to our

knowledge of the different sources of these molecules. While the story is complete in what it set out to demonstrate, I have a few conceptual and technical questions.

Questions:

Microglia express IL-1a in the brain of *T. gondii* infected mice. IL-1a signals via IL-1R on endothelial cells to upregulate VCAM-1 and ICAM-1, which in turn recruit microglia to the brain. Chicken and egg question: Which comes first, IL-1a production in the brain by another cell type and then the amplification loop of IL-1a driven by recruited microglia or microglia recruited independently of IL-1a signalling, which then produce IL-1a leading to more microglia in the brain? In essence is there IL-1a production in the brain without microglia?

What about IL-18, do the authors see any production of IL-18 in their system?

The authors need to better discuss their findings in light of previously published work analysing the importance of casp1/11 and IL-1R during the acute phase of *T. gondii* infection (Gorfu et al). In the published study, these mice (esp. casp1/11KO) mostly do not make it to the chronic phase. I know the parasite stage used for infection is different, is that the reason why there is an observed difference in susceptibility? On the same note, there are >2000 cysts in the brains of their casp1/11KO mice – how do the mice survive that? Are they exhibiting signs of disease at that point?

IL-1a processing is not dependent on inflammatory caspases. By studying gasderminD KO mice, the authors find that IL-1a release is dependent on inflammasome activation/pore formation. However, they do not analyse IL-1a levels in casp1/11KO mice. Presumably these would also be lower as casp1/11 processes gasdermin to form the pore? Is this the case?

Minor comments:

Figure S1b: + missing next to IL-1

Figure 1: In the IF images, why is CX3CR1 labelled differently between panels e and f versus j and l?

Figure 2 onwards in paper: Why are some experiments shown as paired averages comparing groups and others presumably are single points per mouse pooled from all experiments?

P14 top: "cells" missing from sentence

P18: citation for Gorfu et al is missing

Reviewer #1 (*T.gondii* neuro immunology)(Remarks to the Author):

Overview of Manuscript: The manuscript addresses the specific roles of microglia, in infections in the CNS, using the neurotrophic pathogen, *Toxoplasma gondii*, which establishes a chronic infection in the brain, as an experimental model. To identify the specific roles of microglia, the authors used microglia reporter mice, sorted microglia from infiltrating myeloid cells, and analyzed microglia and macrophages using RNA sequencing. Using this approach it was found that NF- κ B and inflammatory cytokines were overly expressed in macrophages vs. microglia and that IL-1 α was enriched in microglia while IL-1 β was enriched in macrophages. To probe this initial finding further the authors used a combination of knockout mice and chemical inhibition. Major findings were: 1) mice lacking IL-1R and IL-1 α but not IL-1 β , have impaired parasite control and immune cell infiltration, 2) microglia release IL-1 α via a gasdermin-D-dependent mechanism and 3) gasdermin-D and caspase 11 deficient mice show deficits in parasite control and immune cell infiltration. The authors conclude macrophages and microglia are differentially equipped to propagate inflammation and that in chronic *T. gondii* infection, microglia release IL-1 α which promotes neuroinflammation and parasite control.

The manuscript aim to define the specific role of microglia is of importance as the role of this resident immune cell in the CNS is not well understood. The main findings of this study that microglia and macrophages are different in their abilities to propagate inflammation in the brain and that microglia release of IL-1 α contributes to control of the chronic *T. gondii* infection, contributes both to the understanding of immune responses to infections in the CNS in general as well as to the understanding of the intracerebral immune response against Toxoplasmosis and will be of interest to the research community in these fields. This is a well-conducted study using cleverly designed experimental approaches with appropriate experimental controls. It is generally well written with the exception of a few issues regarding referencing and an over generalization or lack of understanding of literature and previous work on immune responses to *T. gondii* in the CNS, as detailed below.

Specific Comments:

Introduction

p. 3, 3rd line – the authors state ‘As the only resident immune cell, microglia’. However, astrocytes can also serve as a resident immune effector cell in the brain...’ Furthermore, as studies show astrocytes are an important resident immune effector cell controlling *T. gondii* in the brain, some mention of astrocytes in this first paragraph should be given for proper and complete context (see works by Hunter 2016, 2004; Drögemüller et al., 2008, and others).

We have changed our statement to refer to microglia as “the resident macrophage” in the brain rather than the “only resident immune cell”. We have also included information about the astrocyte response to *T. gondii* infection in the second paragraph, where we feel it fits well into the discussion of the chronic immune response in the brain. We define microglia as an immune cell based on ontogeny, since they derive from the yolk sac (Ginhoux 2010 *Science*) whereas astrocytes derive from the neuroepithelium. Therefore, while astrocytes absolutely have the capacity to contribute to an immune response, the same can be said for many other cell types including fibroblasts, endothelial cells, epithelial cells, etc., which are not typically defined as immune cells. Nevertheless, we absolutely agree that astrocytes are a relevant player and have included astrocytes in our discussion of the immune response (page 3, 2nd paragraph).

p. 3, 2nd paragraph, 3rd sentence: The authors state in reference to the immune response to *T. gondii* “T-cell derived IFN- γ is one essential element” and cite 2 references. The literature on the immune response to *T. gondii* is voluminous and this simple statement referring to IFN- γ , as ‘one’ essential element is misleading about the role of IFN- γ . IFN- γ is ‘the’ essential cytokine controlling *T. gondii* in the brain with its role well defined by work by Y. Suzuki and numerous others. Additionally there is a wealth of knowledge of the immune response to this parasite in the brain. The work on the importance of IFN- γ in control of *T. gondii* by Suzuki should be cited as well as a more complete summary of the large body of knowledge known of the intracerebral immune response to *T. gondii*.

Again, later in this paragraph the authors state the role of microglia in the chronic infection has not been fully ‘elucidated’ which is true, but Suzuki has done significant work on microglia in the acute phase (2007, 2005) which should be cited and referenced for broader context of what is known about the role of microglia in Toxoplasmosis in the brain. As the work presented in this manuscript primarily addresses the role of microglia in the chronic phase of the immune response to *T. gondii* and that aspect could be stated in the above statement to discriminate from what is known of microglia in the acute infection.

Results

p. 5, 1st paragraph – The statement “Blood-derived monocytes have been demonstrated to be important for host survival during infection, but whether microglia perform similar functions is still unknown” seems to be an overstatement and in part wrong. For example Suzuki’s work on microglia has shown that microglia are important as sentinel cells, able to detect early cerebral *Toxoplasma* proliferation and activate their IFN- γ production thus facilitating protection immunity in controlling parasite proliferation in the brain. This is essentially the same issue as noted above in the Introduction.

Discussion

p. 21 – 1st paragraph – The statement ‘Much is known about the immune response to this parasite but the role of the brain-resident microglia is still largely unknown.’ again reads as a little untrue or at least too broadly stated as the role of microglia in IFN- γ mediated protection as mentioned above is well established, and more recent evidence indicating migratory properties of infected microglia possibly serve to disseminate the parasite in the brain, are also in the literature. Hence this statement is just too general - a qualification of ‘in the chronic infection’ or some type of delineating/restricting condition, could be used to help clarify what is new in this manuscript while also indicating what is known of the understanding of microglia in Toxoplasmosis.

Later in this paragraph the authors discuss the in vitro experiments done with *T. gondii* infected microglia and discuss how in vitro systems don’t adequately reflect the biology/phenotype of microglia in vivo. These in vitro studies are quite old and since that time most studies addressing the role of microglia have been done in vivo so this statement seems to ignore or at least erroneously suggest the in vitro studies are the main source of information on microglia and *Toxoplasma*.

We thank the reviewer for reminding us of literature that assigns a role to microglia during *T. gondii* infection. Accordingly, we have made changes to the Introduction, Results, and Discussion sections which have been highlighted in the main text (pages 3, 4, 6, 22, 23).

As a concluding statement to the authors, I wish to express that despite the above comments, I believe this is a well-done study is well-done that addresses an important question about the

differential roles of macrophages and microglia in the brain in *T. gondii* which hasn't received attention before. However the seeming lack of knowledge and/or understanding of the literature on work done on the role of microglia in the cerebral immune response to *T. gondii* bothered me throughout the manuscript. Nearly, all the specific comments refer to this issue, which occurred in several sections of the manuscript. However, addressing this issue I believe, will made this a better paper but also will put a greater emphasis on that in the manuscript that is truly new and noteworthy, and thus significantly improve the manuscript.

Reviewer #2 (*T.gondii* Immune responses)(Remarks to the Author):

Batisda et al follow a logical approach to delineate a molecular function for microglia in control of chronic *T. gondii* infection in the brain. They discover that rather than the expected NFkB transcriptional signature present in infiltrating macrophages, brain microglia are skewed to express IL-1a. These findings are corroborated on protein level. Mice lacking IL-1R1 and IL-1a, but not IL-1b, exhibit impaired parasite control and immune cell infiltration into the brain in the chronic phase. Microglia ex vivo are shown to produce IL-1a, dependent on gasdermin D. Gasdermin D and caspase 1/11 knockout mice equally present with decreased parasite control and defect immune cell infiltration into the brain.

This study addresses important differences between macrophages and microglia in controlling chronic *T. gondii* by profiling their inflammatory potential in the brain. IL-1a is the “neglected” IL-1 in the context of inflammasome activation and IL-1R signalling, and this study adds to our knowledge of the different sources of these molecules. While the story is complete in what it set out to demonstrate, I have a few conceptual and technical questions.

Questions:

Microglia express IL-1a in the brain of *T. gondii* infected mice. IL-1a signals via IL-1R on endothelial cells to upregulate VCAM-1 and ICAM-1, which in turn recruit microglia to the brain. Chicken and egg question: Which comes first, IL-1a production in the brain by another cell type and then the amplification loop of IL-1a driven by recruited microglia or microglia recruited independently of IL-1a signalling, which then produce IL-1a leading to more microglia in the brain? In essence is there IL-1a production in the brain without microglia?

We apologize that this was unclear in the manuscript. IL-1 α is expressed by microglia in the brain at baseline prior to infection (Fig. S1). Infection of the brain occurs before substantial immune infiltration to the brain starts to kick in. Infection leads to release of IL-1 α which, in concert with many other factors like IFN- γ , activates the endothelium and promotes the infiltration of monocytes and monocyte-derived macrophages from the blood. Though these recruited cells may look like microglia, they never fully become microglia, as ontogeny seems to be important (including for IL-1 β production which we observe only in macrophages). Further, the recruited monocytes/macrophages express IL-1 α , but our data show that only the microglia release it (Fig. 5).

What about IL-18, do the authors see any production of IL-18 in their system?

This is an interesting question given the results seen in the Gofu et al. study. Though it is another IL-1 family cytokine, our results regarding the importance of IL-1 in this model would not be confounded by IL-18 because it does not share the same receptor as IL-1 α and IL-1 β (IL-

1R1). However, as a matter of interest in what could be another relevant pathway, we assessed RNA expression of *Il18* in whole brain homogenate in naïve and infected samples. Results are shown below (A). Though there is a statistically significant increase in *Il18* expression the magnitude of this change is small (A). We have included the change in *Il1a* expression in these same samples as a reference (B). Probably more importantly, we also assessed IL-18 release *ex vivo* from cells isolated from chronically infected brain samples (C). No IL-18 release was detected in either control infected samples (cellular release of cytokine) or samples treated with a triton-containing lysis buffer (measurement of cytokine from lysed cells). This suggests that is likely not present in the samples. Moreover, the RNAseq analyses performed on microglia and macrophages indicated that both cell types express no to extremely low levels of IL-18. Thus, we do not have evidence that supports microglia or macrophage release of IL-18.

The authors need to better discuss their findings in light of previously published work analysing the importance of casp1/11 and IL-1R during the acute phase of *T. gondii* infection (Gorfu et al). In the published study, these mice (esp. casp1/11KO) mostly do not make it to the chronic phase. I know the parasite stage used for infection is different, is that the reason why there is an observed difference in susceptibility? On the same note, there are >2000 cysts in the brains of their casp1/11KO mice – how do the mice survive that? Are they exhibiting signs of disease at that point?

This is an interesting point that we have given a lot of thought. Because of the results seen in the Gorfu et al. study, we were very surprised that in our hands the IL-1R and caspase-1/11 deficient mice survived to chronic infection. Different parasite strains and dose (10 cysts vs. 10,000 tachyzoites; Me49 vs. 76K GFP-LUC) and different conditions in the vivaria could potentially contribute to these differences. We would also like to note that in the Gorfu et al. paper a large proportion of the WT mice succumb to infection as well, which is not something that we observe, as we aim to model chronic asymptomatic infection that occurs in nearly all hosts, including humans. As for the cyst load in the brain, this is very interesting because the mice are actually able to tolerate that amount of parasite without exhibiting morbidity. In fact, we find that they are able to harbor over 10,000 cysts without showing signs of disease. This may indicate that it is the replicating form of the parasite rather than the cysts form that causes more damage and morbidity.

IL-1a processing is not dependent on inflammatory caspases. By studying gasderminD KO mice, the authors find that IL-1a release is dependent on inflammasome activation/pore formation. However,

they do not analyse IL-1 α levels in casp1/11KO mice. Presumably these would also be lower as casp1/11 processes gasdermin to form the pore? Is this the case?

We did not measure IL-1 α release in the caspase-1/11 KO mice; we focused more on using the gasdermin-D KO mice because that allows us to point specifically to gasdermin pore formation as a mechanism of IL-1 α release. We would have liked to have addressed your question directly, but with the shutdown of research activity at the University of Virginia, we are unable to assess this at this time. Like you, we hypothesize that gasdermin would be processed by either caspase 1 or 11 as has been demonstrated in the literature. We do have preliminary data showing that *ex vivo* IL-1 α release is not dependent on caspase-11 (below), and thus our hypothesis is that it will be dependent on caspase-1, which would make sense with the ASC specks we see in infected brains (Figure 6). Since we do not currently have the data to support this however, we describe this process as gasdermin-D-dependent, rather than inflammasome-dependent. It should also be noted that there are other pathways being discovered through which gasdermin D can be cleaved, including by caspase-8 (Sarhan 2018 *PNAS*). In the future we plan to explore some of these possible pathways.

Minor comments:

Figure S1b: + missing next to IL-1

Figure 1: In the IF images, why is CX3CR1 labelled differently between panels e and f versus j and l?

Figure 2 onwards in paper: Why are some experiments shown as paired averages comparing groups and others presumably are single points per mouse pooled from all experiments?

P14 top: "cells" missing from sentence

P18: citation for Gorfu et al is missing

Thank you for catching these things which we missed. They have all been corrected accordingly. In regards to the paired averages, some experiments in this work were performed more than three times. And while we wanted to show the power and consistency in the results by including data points from many separate experiments, the number of data points began to overwhelm the plots, obfuscating the results. But when there were a more manageable number of points, we wanted to include the data for each individual mouse, in the interest of transparency. To that end, the raw data contributing to the paired averages will be provided in a source data file as requested by the journal.

REVIEWERS' COMMENTS:

Reviewer #1 (Remarks to the Author):

The authors have satisfactorily addressed the reviewers concerns. This is a much improved manuscript that presents important new information. This manuscript contains new insights into the differential roles of microglia and macrophages to propagate inflammation and contributes new information on the roles of microglia and the alarmin IL-1a, to promote neuroinflammation and parasite control in the brain. This information will be of interest to the parasite community and contributes to the understanding of the immune response to *Toxoplasma gondii* in the brain.

Reviewer #2 (Remarks to the Author):

Thank you for addressing all of my questions raised and making the corrections requested.

Please find a response to reviewers' comments below. Our response is in blue text.

REVIEWERS' COMMENTS:

Reviewer #1 (Remarks to the Author):

The authors have satisfactorily addressed the reviewers concerns. This is a much improved manuscript that presents important new information. This manuscript contains new insights into the differential roles of microglia and macrophages to propagate inflammation and contributes new information on the roles of microglia and the alarmin IL-1a, to promote neuroinflammation and parasite control in the brain. This information will be of interest to the parasite community and contributes to the understanding of the immune response to *Toxoplasma gondii* in the brain.

Reviewer #2 (Remarks to the Author):

Thank you for addressing all of my questions raised and making the corrections requested.

We thank the reviewers for their comments on our original submission and we are glad we have addressed all of your questions.

Sincerely,

Tajie H. Harris, PhD
Associate Professor of Neuroscience
tajieharris@virginia.edu